# The role of polygenic susceptibility to obesity among carriers of pathogenic mutations in *MC4R* in the UK Biobank population

Nathalie Chami[1,2], Michael Preuss[1,2], Ryan W. Walker[3], Arden Moscati[1], Ruth J. F. Loos[1,2,3]*

1 The Charles Bronfman Institute for Personalized Medicine, Icahn School of Medicine at Mount Sinai, New York, New York, United States of America, 2 The Mindich Child Health and Development Institute, Icahn School of Medicine at Mount Sinai, New York, New York, United States of America, 3 Department of Environmental Medicine and Public Health, Icahn School of Medicine at Mount Sinai, New York, New York, United States of America

* Ruth.loos@mssm.edu

## Abstract

### Background

Melanocortin 4 receptor (MC4R) deficiency, caused by mutations in *MC4R*, is the most common cause of monogenic forms of obesity. However, these mutations have often been identified in small-scale, case-focused studies. Here, we assess the penetrance of previously reported *MC4R* mutations at a population level. Furthermore, we examine why some carriers of pathogenic mutations remain of normal weight, to gain insight into the mechanisms that control body weight.

### Methods and findings

We identified 59 known obesity-increasing mutations in *MC4R* from the Human Gene Mutation Database (HGMD) and Clinvar. We assessed their penetrance and effect on obesity (body mass index [BMI] $\geq$ 30 kg/m$^2$) in >450,000 individuals (age 40–69 years) of the UK Biobank, a population-based cohort study. Of these 59 mutations, only 11 had moderate-to-high penetrance and increased the odds of obesity by more than 2-fold.

We subsequently focused on these 11 mutations and examined differences between carriers of normal weight and carriers with obesity. Twenty-eight of the 182 carriers of these 11 mutations were of normal weight. Body composition of carriers of normal weight was similar to noncarriers of normal weight, whereas among individuals with obesity, carriers had a somewhat higher BMI than noncarriers (1.44 ± 0.07 standard deviation scores [SDSs] ± standard error [SE] versus 1.29 ± 0.001, *P* = 0.03), because of greater lean mass (1.44 ± 0.09 versus 1.15 ± 0.002, *P* = 0.002). Carriers of normal weight more often reported that, already at age 10 years, their body size was below average or average (72%) compared with carriers with obesity (48%) (*P* = 0.01).

To assess the polygenic contribution to body weight in carriers of normal weight and carriers with obesity, we calculated a genome-wide polygenic risk score for BMI (PRS$_{BMI}$). The

**Data Availability Statement:** Data of the UK Biobank can be obtained directly from the UK Biobank (http://biobank.ndph.ox.ac.uk). Details on

the application process are described here https://www.ukbiobank.ac.uk/researchers/.

**Funding:** This research was supported by the National Institutes of Health (R01DK110113; R01DK124097) and by an Alliance Award of the University of Copenhagen (Denmark), NNF Center for Basic Metabolic Research. NC is supported by a grant from the Canadian Institutes of Health Research (CIHR Fellowship). The funders had no role in study design, data collection and analysis, decision to publish, or preparation of the manuscript.

**Competing interests:** The authors have declared that no competing interests exist.

**Abbreviations:** BF%, body fat percentage; BMI, body mass index; FFMI, fat-free mass index; GoF, gain-of-function; GWAS, genome-wide association study; HGMD, Human Gene Mutation Database; IPAQ, International Physical Activity Questionnaire; LoF, loss-of-function; MAF, minor allele frequency; MC4R, melanocortin 4 receptor; MET, metabolic equivalent minute; OR, odds ratio; PC, principle component; PRS, polygenic risk score; PRS$_{BMI}$, polygenic risk score for BMI; PRS-SD, polygenic risk score standard deviation; SD, standard deviation; SDS, standard deviation score; SE, standard error; STROBE, Strengthening The Reporting of OBservational Studies in Epidemiology; TDI, Townsend Deprivation Index; WHR, waist-to-hip ratio.

PRS$_{BMI}$ of carriers of normal weight (PRS$_{BMI}$ = -0.64 ± 0.18) was significantly lower than of carriers with obesity (0.40 ± 0.11; $P$ = 1.7 × 10$^{-6}$), and tended to be lower than that of non-carriers of normal weight (−0.29 ± 0.003; $P$ = 0.05). Among carriers, those with a low PRS$_{BMI}$ (bottom quartile) have an approximately 5-kg/m$^2$ lower BMI (approximately 14 kg of body weight for a 1.7-m-tall person) than those with a high PRS (top quartile).

Because the UK Biobank population is healthier than the general population in the United Kingdom, penetrance may have been somewhat underestimated.

## Conclusions

We showed that large-scale data are needed to validate the impact of mutations observed in small-scale and case-focused studies. Furthermore, we observed that despite the key role of *MC4R* in obesity, the effects of pathogenic *MC4R* mutations may be countered, at least in part, by a low polygenic risk potentially representing other innate mechanisms implicated in body weight regulation.

## Author summary

### Why was this study done?

- Obesity is a major risk factor for type 2 diabetes, cardiovascular disease, chronic kidney disease, and many cancers.

- The melanocortin 4 receptor (*MC4R*) plays an important role in regulating energy balance and satiety. Mutations in *MC4R*, although rare (<1% of the population), represent the commonest cause of extreme early onset obesity.

- Mutations in *MC4R* have been identified predominantly in small-scale studies of individuals with obesity. The mutations' impact on obesity risk in the general population remains to be studied. Furthermore, it is not clear why some carriers of *MC4R* mutations maintain a normal weight, even when the mutation was shown to increase risk of obesity.

### What did the researchers do and find?

- For 59 mutations previously reported to possibly cause obesity, we determined how many individuals, of a large-scale, population-based cohort ($N$ > 450,000), carried the obesity-increasing allele and how many of these carriers had obesity (i.e., penetrance of mutation).
  For 11 of these mutations, the penetrance of obesity was high.

- Of the 182 individuals who carried at least one of these 11 mutations, 154 (85%) individuals had obesity/overweight, whereas 28 (15%) individuals were of normal weight.

- We observed that, compared with carriers who had obesity, the 28 carriers of normal weight have other inherited genetic variants that overall predispose them to a lower body weight, which may offset the risk caused by the *MC4R* mutation they carry.

**What do these findings mean?**

- Our findings show that large-scale population data are needed to more accurately assess the impact of *MC4R* mutations on extreme early onset obesity.

- We show that the obesity-increasing effect of *MC4R* mutations may be mitigated by a low overall genetic susceptibility to obesity.

- These findings show that body weight is the result of an intricate interplay between rare mutations that have a large impact on obesity, as well as an overall genetic susceptibility determined by common genetic variants each with small effects.

## Introduction

Obesity is a major risk factor for leading causes of mortality, including type 2 diabetes, cardio-vascular disease, chronic kidney disease, and many cancers [1]. To date, more than 650 million adults worldwide suffer from obesity (body mass index [BMI] $\geq 30$ kg/m$^2$), a tripling over the past 4 decades [2]. Obesity is the result of an intricate interplay between genetic susceptibility and an obesogenic environment. In a fraction of cases, obesity is caused by mutations in a single gene, resulting in severe early onset obesity. In these instances, environment is believed to play only a minimal role. Many of the pathogenic mutations that cause severe early onset obesity affect genes and proteins in the leptin-melanocortin pathways (e.g., *LEP*, *LEPR*, *POMC*, *PCSK1*, and *MC4R*) [3,4].

Mutations in the melanocortin 4 receptor (*MC4R)* represent the commonest cause of severe early onset obesity [5]. It has been estimated that up to 5% of patients with severe childhood obesity carry pathogenic mutations that cause MC4R deficiency [6,7]. Patients with MC4R deficiency exhibit hyperphagia from an early age [6,7]. Besides increased fat mass, they also have more lean mass, greater bone-mineral density, and are taller compared with non-*MC4R* deficient individuals with obesity [8]. The severity of the clinical phenotype varies, depending on the functional implications of the mutation on the receptor [6].

In a recent large-scale genetic association study, a well-known nonsense mutation (p.Tyr35Ter, rs13447324) in *MC4R* was associated with approximately 7-kg higher body weight in carriers (approximately 1 in 5,000 people) [9]. This mutation has been repeatedly found to be the cause of severe early onset obesity [5,6,10–13]. In-depth functional analyses show that Tyr35Ter results in a complete loss-of-function (LoF) of *MC4R* [12,14–17]. Nevertheless, of the 30 mutation carriers identified in approximately 120,000 participants of the UK Biobank, 6 (20%) were of normal weight [9]. This observation supports the notion that penetrance of pathogenic *MC4R* mutations is incomplete and that genetic and/or nongenetic factors may affect the clinical outcome. Owing to the fact that *MC4R* mutations have typically been identified through small-scale and case-focused studies, the estimates of penetrance and representation of clinical phenotypes are likely to be biased because of ascertainment. With the availability of large-scale unselected population datasets with extensive genotype and phenotype data, it is now possible to assess the penetrance at the population level.

Here, we first assess the impact of pathogenic *MC4R* mutations, previously implicated in severe and early onset obesity, in the UK Biobank, a large-scale population-based cohort of approximately 500,000 individuals living in the United Kingdom. Next, we examine why some individuals who carry these *MC4R* mutations are able to remain of normal weight. These observations may provide new insights into the mechanisms that control body weight.

## Methods

### Study population

All analyses are based on data from the UK Biobank, a prospective cohort study with extensive genetic and phenotypic data collected in approximately 500,000 individuals, aged between 40–69 years. Participants were enrolled from April 2007 to July 2010 at one of 21 assessment centers across the UK to provide baseline information, physical measures, and biological samples according to standardized procedures [18–21]. Questionnaires were used to collect health and lifestyle data [22]. Study design, protocols, sample handling and quality control have been described in detail elsewhere [18–21].

We restricted analyses to individuals of European ancestry (*N* = 453,800) (S1 Text). Women who were pregnant at the time of recruitment (*N* = 119) and individuals with poor-quality samples based on metrics for heterozygosity or missingness (*N* = 421) were excluded. Individuals (*N* = 274) who underwent weight loss surgery before recruitment were considered as "individuals with obesity" for the penetrance calculation. They were removed for the subsequent analyses leaving 452,986 individuals (207,350 men and 245,636 women) of European ancestry in the study.

The UK Biobank received ethical approval from the North West–Haydock Research Ethics Committee (REC reference 11/NW/0382). The current study was conducted under UK Biobank application 1251. Appropriate informed consent was obtained from the participants.

### Phenotype data

All phenotypic data used for analyses were collected at the baseline visit. We provide a brief description here; more details can be found in S1 Text and elsewhere [18–21]. BMI, calculated as weight (kg) divided by height squared (m$^2$), was used to categorize individuals with underweight (BMI < 18.5 kg/m$^2$), normal weight (18.5 kg/m$^2$ $\leq$ BMI < 25 kg/m$^2$), overweight (25 kg/m$^2$ $\leq$ BMI < 30 kg/m$^2$), or obesity (BMI $\geq$ 30 kg/m$^2$). Waist-to-hip ratio (WHR) was calculated by dividing waist over hip circumference. Body composition (fat mass and fat-free mass) and basal metabolic rate was assessed by bioimpedance. Body fat percentage (BF%) was calculated as fat mass (kg) divided by weight (kg) times 100. Fat-free mass index (FFMI) was calculated as fat-free mass (kg) divided by height squared (m$^2$) to assess relative leanness [23].

Birth weight (kg), participants' "comparative body size at age 10" and "comparative height at age 10", and age at menarche (years) for women were obtained through self-report. Daily physical activity was assessed using the International Physical Activity Questionnaire (IPAQ), which allows the categorization of individuals into 3 groups of physical activity levels (low, moderate, and high). In addition to IPAQ, we also used the summed MET-minute/week for all activities as a continuous trait. Quantitative information on diet was not available for sufficient participants (in particular, mutation carriers) to allow analyses. The Townsend Deprivation Index (TDI) was used as a proxy of socioeconomic status; a negative value represents high socioeconomic position [24]. Descriptive characteristics are presented in S1 Table.

### Genotype data

The majority (approximately 450,000) of UK Biobank participants were genotyped using the UK Biobank Axiom Array, a custom-designed array that includes 821,000 genetic markers. A subset of participants (approximately 50,000) was genotyped using the UK BiLEVE Array, which was designed first, with the aim to study the genetics of lung health and disease, and has >95% markers in common with the UK Biobank Axiom Array. Variant- and sample-based quality control was performed by the UK Biobank; including testing for batch, plate, and array

effects, Hardy-Weinberg equilibrium, and discordance across control replicates. Poor-quality samples with high missingness rate and heterozygosity were identified using high-quality markers from both arrays [18]. Because imputation quality of rare variants (minor allele frequency [MAF] $\leq$ 0.1%) is generally low, only genotyped variants were analyzed.

Genotyped data were used to identify individuals of European ancestry, as described in S1 Text.

## Mutations in *MC4R* reported to play a role in obesity

We used the Human Gene Mutation Database (HGMD) [25] and Clinvar (a clinical genetic database) [26] databases to collate all mutations (MAF $\leq$ 0.1%) in *MC4R* that have been reported previously as having a role in obesity.

The HGMD, a database of mutations that have been reported to be associated with inherited diseases, reported 150 mutations in *MC4R* (accessed November 2018). Of these 150 mutations, 74 had been genotyped in the UK Biobank, of which we excluded 4 mutations because of low call rate (<90%) and 1 because of high MAF (MAF = 1.9%). Thus, a total of 69 mutations reported by HGMD were retained. In addition to HGMD, ClinVar reported 51 mutations in *MC4R* with a role in obesity, of which 20 were genotyped in the UK Biobank, all of which were also listed in HGMD.

## Quality control of *MC4R* variants

We assessed the quality of the genotype data for each of the 69 mutations following the procedures proposed by UK Biobank Access Team [27]. They made the following recommendations, based on quality control procedures reported by others [18,28,29]

Specifically, we examined the individual cluster plots for each of the 69 variants and assessed the concordance and discordance between the genotyped and sequenced data available in approximately 10% of the UK Biobank participants of European ancestry ($N$ = approximately 46,000) (S1 Text). As such, we identified 10 mutations of low quality (cluster plots were of "poor" or "intermediate" quality) (S1 Text, S2 Table) that were removed from downstream analyses. We also "flagged" 20 additional variants that were not fully concordant between the genotyped and the sequenced data. Given that the sequenced subset consists of only approximately 10% of the full population analyzed, we chose to not remove these mutations from our analyses (S1 Text, S2 Table). Removing the high-impact variants that were flagged in a sensitivity analysis does not impact our main conclusions (S1 Text, S1 and S3 Tables, S1 and S2 Figs).

Taken together, 59 *MC4R* mutations that had previously been reported to play a role in obesity were included in our analyses.

## Assessing the impact of identified *MC4R* mutations in obesity among UK Biobank participants

Although HGMD and ClinVar are valuable collections of reported mutations, they do not intend to validate their functional implications or role in disease. Furthermore, the studies that have identified *MC4R* mutations are typically small and case focused. To confirm the mutations' impact on obesity, we first determined their penetrance and effect among the 452,198 individuals of the UK Biobank (Fig 1).

For each mutation, we calculated the penetrance of obesity as the percentage of individuals with obesity among carriers of a given mutation (= [# carriers with obesity/total # of carriers] × 100). In addition, we calculated the odds of mutation carriers to have obesity compared with them being of normal weight and the odds of noncarriers to have obesity compared with them being of normal weight to derive odds ratios (ORs) for each mutation. Because there are

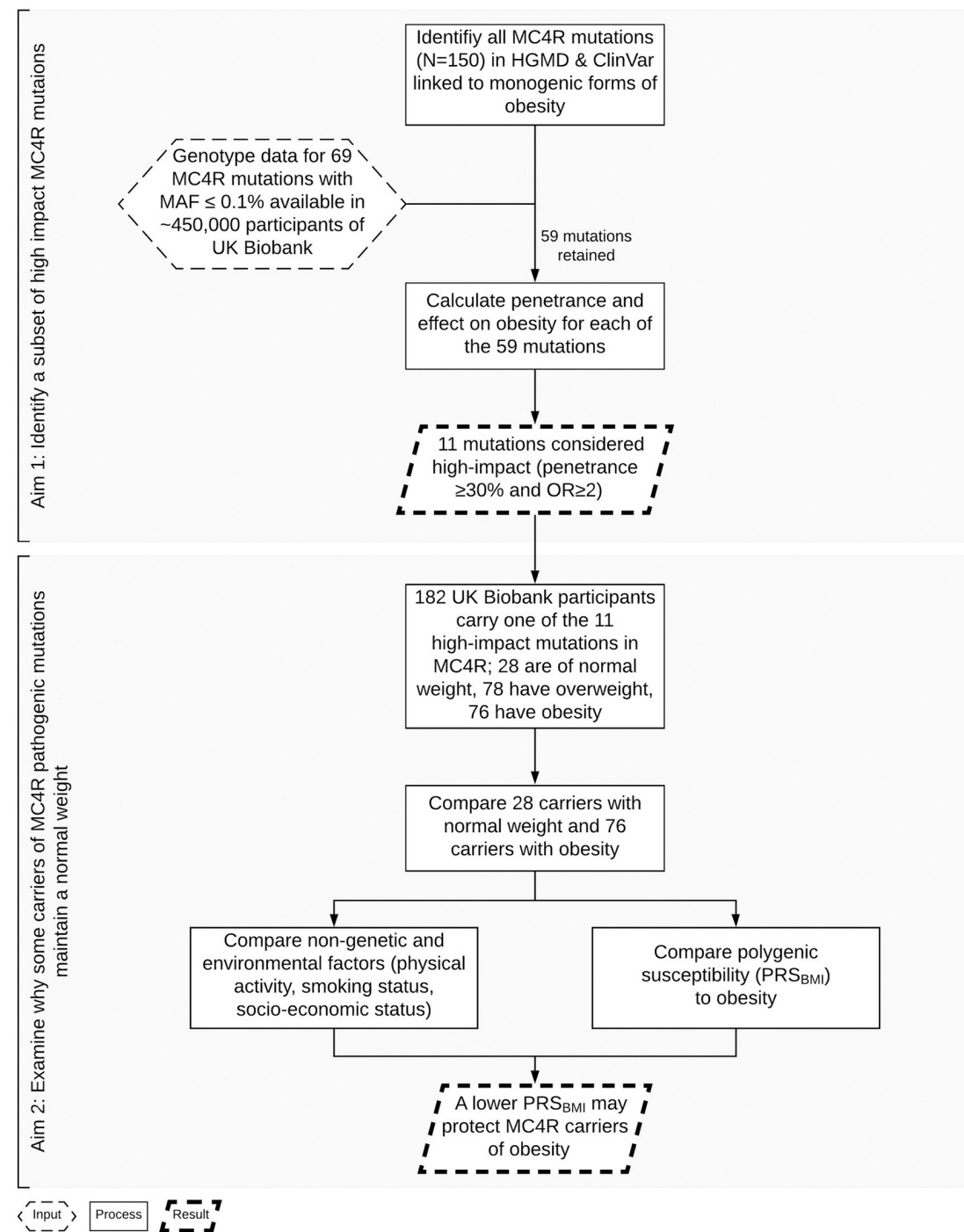

**Fig 1. Flow chart of the study design.** HGMD, Human Gene Mutation Database; MAF, minor allele frequency; MC4R, melanocortin 4 receptor; OR, odds ratio.

mutations for which there were no carriers of normal weight, we added a constant ("+1") to the number of carriers with obesity (numerator) and to the number of carriers of normal weight (denominator) to allow deriving "proxy" ORs. These ORs are only used to identify "high-impact" mutations and not to quantify the exact effect of the mutation on obesity as

such. The combination of penetrance and effect (OR) allows the identification of mutations that likely have the biggest impact on obesity risk (Fig 2, S1 Text).

The mean penetrance and OR across the 11 high-impact mutations was calculated using individual level data, rather than the average across the 11 values.

### Polygenic risk score for BMI

A polygenic risk score (PRS) assesses a person's overall genetic susceptibility to a certain trait or disease. We calculated a PRS for BMI ($PRS_{BMI}$) using the software PRSice (https://www.prsice.info) [30] and summary statistics from the most recent genome-wide association study (GWAS) meta-analyses for BMI [31] that does not include data from the UK Biobank. Our $PRS_{BMI}$ included 351,597 variants and was, as expected, significantly associated with BMI (1.29 kg/m$^2$ per PRS-standard deviation [PRS-SD], $P < 2 \times 10^{-16}$). More details are available in S1 Text.

### Statistical analyses

We created sex-specific residuals for all continuous outcomes, adjusted for age and the first 10 genetic principle components (PCs) using linear regression, followed by inverse normal transformation. As such, transformed traits are on the same scale with a mean of 0 and a standard deviation (SD) of 1, which allows direct comparison of effects sizes across traits. In a secondary analysis, we additionally adjusted for physical activity (MET), smoking behavior, and TDI to compare $PRS_{BMI}$.

We used Welch's t-test for continuous traits and the Fisher's Exact and Cochrane Armitage Trend tests for discrete traits to compare differences between (1) carriers and noncarriers within each BMI category (normal weight and obesity) and between (2) individuals of normal weight and with obesity among carriers and noncarriers.

To quantify the effect of carrier status and polygenic risk on BMI and obesity risk, we assessed the difference between individuals with a low polygenic risk (lowest quartile of PRS) and a high polygenic risk (highest quartile) among *MC4R* mutation carriers and noncarriers. Individuals with a low PRS who were noncarriers were considered to be at the lowest genetic risk and served as the "reference group." This reference group was compared with (1) *MC4R* mutation carriers with a low polygenic risk, (2) noncarriers with a high polygenic risk, and (3) *MC4R* mutation carriers with a high polygenic risk (highest genetic risk). We used logistic regression to calculate OR for obesity comparing each group to the reference group. All analyses were performed using R (https://www.r-project.org) and PLINK v.1.9 (https://www.cog-genomics.org/plink2) [32].

This study is reported as per the Strengthening The Reporting of OBservational Studies in Epidemiology (STROBE) guideline (S1 Checklist).

## Results

### *MC4R* mutations reported to cause obesity have variable penetrance and effect on obesity

We studied 69 mutations, previously reported to cause obesity in small-scale and case-focused studies, in 452,128 individuals of the UK Biobank (Table 1, S1 Table). Of these, 59 mutations passed quality control criteria (Methods, S2 Table, S1 Text). Their penetrance ranged from 0% for 13 mutations to 100% for 2 mutations, with an average of 25.3% across all 59 mutations (Fig 2, S4 Table). Their effect (OR) on obesity risk ranged from 0.22 (protective) to 9.43 (risk),

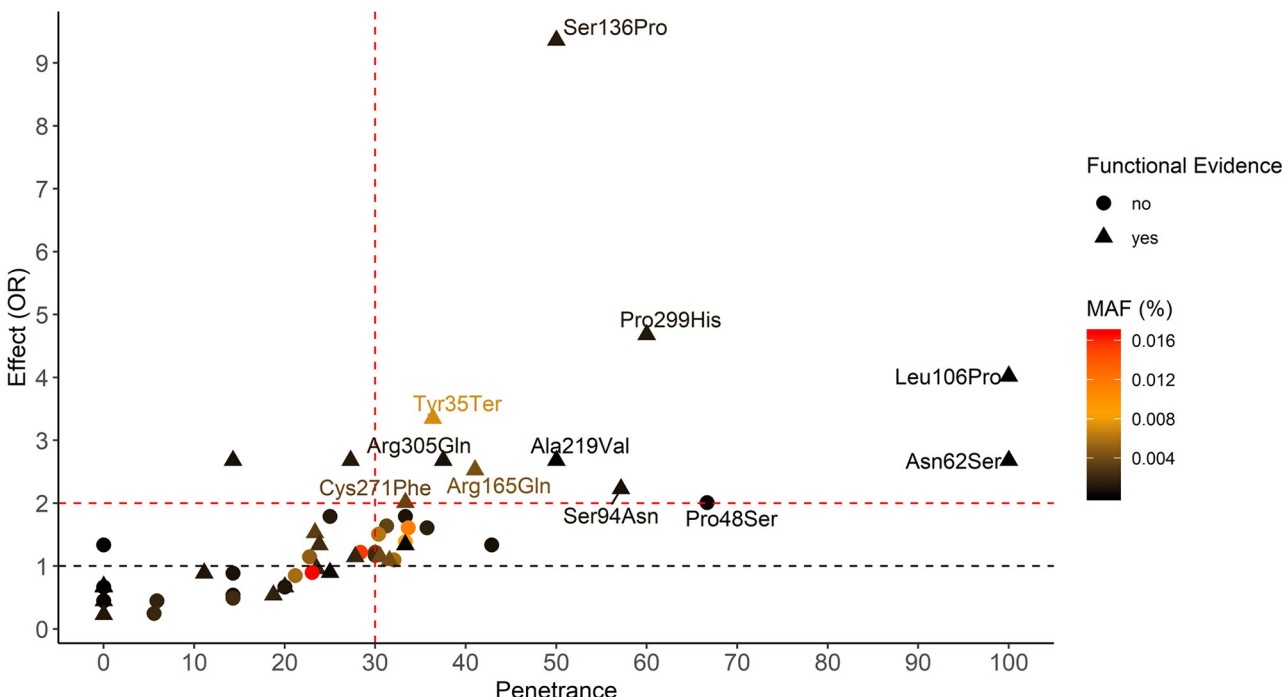

**Fig 2. Penetrance and effect on obesity (OR) of 59 *MC4R* mutations previously reported to cause obesity.** The red dotted lines denote the thresholds of penetrance (≥30%) and effect (OR ≥ 2) that determine impact of mutations. The amino acid change for the top-ranking mutations are labeled. The effect is the odds of mutation carriers to have obesity compared with the odds of them being of normal weight. MAF, minor allele frequency; OR, odds ratio.

with an average OR of 1.22. For 6 mutations (MAF of approximately 0.0001%), we observed only 1 carrier, and for 3 mutations (MAF > 0.01%), there were more than 100 carriers.

## Eleven *MC4R* mutations with high impact on obesity

The combination of penetrance and effect allows identifying the mutations with the biggest impact on obesity. Given that the prevalence of obesity among noncarriers in the UK Biobank is 24%, mutations for which ≥30% of carriers had obesity were considered moderately-to-highly penetrant. As the ratio of obesity over normal weight among noncarriers in the UK Biobank is 0.74 (i.e., more normal weight than obesity), mutations that increase the odds of obesity by 2-fold or more (compared with normal weight) were considered to have a moderate-to-high risk.

Eleven mutations met both penetrance (≥30%) and effect (OR ≥ 2) thresholds and were considered for further analyses (Fig 2; S4 Table). Their average penetrance (42%) and effect on obesity (OR = 3.49) were substantially higher than that of the remaining 48 mutations (26%; OR = 1.04). For 10 (91%) of the 11 mutations, there was evidence that the mutation impaired MC4R function and/or led to reduced activity, based on functional characterization (S4 and S5 Tables). In contrast, functional evidence was reported for only 20 (42%) of the remaining 48 mutations ($P = 0.006$).

## Carriers of high-impact *MC4R* mutations have higher BMI and have more often obesity

There were 183 individuals (0.004% or 1 in 2,471 individuals) who carried one of the 11 mutations. No individuals carried more than 1 mutation, and all were heterozygous carriers. One

**Table 1. Comparison between carriers and noncarriers of normal weight and with obesity (in adjusted standardized scores).**

| Variable | BMI category | Carriers N | Carriers Mean ± SE | Carriers P value (normal weight versus obesity) | Noncarriers N | Noncarriers Mean ± SE | Noncarriers P value (normal weight versus obesity) | P value (carrier versus noncarrier) |
|---|---|---|---|---|---|---|---|---|
| Age | Normal weight | 28 | −0.14 ± 0.15 | 0.40 | 147,530 | −0.10 ± 0.003 | $<2\times10^{-16}$ | 0.80 |
| | Obesity | 76 | 0.02 ± 0.11 | | 109,178 | 0.04 ± 0.003 | | 0.90 |
| BMI | Normal weight | 28 | −0.98 ± 0.12 | $<2\times10^{-16}$ | 147,530 | −1.04 ± 0.001 | $<2\times10^{-16}$ | 0.60 |
| | Obesity | 76 | 1.44 ± 0.07 | | 109,178 | 1.29 ± 0.001 | | 0.03 |
| WHR | Normal weight | 28 | −0.46 ± 0.18 | $2.3\times10^{-6}$ | 147,514 | −0.59 ± 0.002 | $<2\times10^{-16}$ | 0.50 |
| | Obesity | 76 | 0.64 ± 0.10 | | 109,116 | 0.73 ± 0.003 | | 0.40 |
| Body fat percentage | Normal weight | 27 | −0.8 ± 0.13 | $2.6\times10^{-16}$ | 145,396 | −0.88 ± 0.002 | $<2\times10^{-16}$ | 0.50 |
| | Obesity | 74 | 1.10 ± 0.10 | | 107,316 | 1.13 ± 0.002 | | 0.80 |
| FFMI | Normal weight | 27 | −0.86 ± 0.16 | $1.5\times10^{-15}$ | 145,599 | −0.87 ± 0.002 | $<2\times10^{-16}$ | 0.91 |
| | Obesity | 74 | 1.44 ± 0.09 | | 107,273 | 1.15 ± 0.002 | | 0.001 |
| Height | Normal weight | 28 | 0.33 ± 0.21 | 0.30 | 147,530 | 0.11 ± 0.003 | $<2\times10^{-16}$ | 0.30 |
| | Obesity | 76 | 0.07 ± 0.11 | | 109,178 | −0.13 ± 0.003 | | 0.09 |
| Birth weight | Normal weight | 22 | −0.23 ± 0.19 | 0.06 | 87,942 | −0.02 ± 0.003 | $<2\times10^{-16}$ | 0.30 |
| | Obesity | 40 | 0.23 ± 0.13 | | 61,372 | 0.04 ± 0.004 | | 0.20 |
| Age at menarche | Normal weight | 19 | −0.05 ± 0.23 | 0.94 | 96,306 | 0.12 ± 0.003 | $<2\times10^{-16}$ | 0.50 |
| | Obesity | 43 | −0.05 ± 0.16 | | 56,385 | −0.20 ± 0.004 | | 0.30 |
| Townsend Deprivation Index | Normal weight | 28 | −0.18 ± 0.20 | 0.05 | 147,371 | −0.06 ± 0.003 | $<2\times10^{-16}$ | 0.60 |
| | Obesity | 76 | 0.28 ± 0.11 | | 109,033 | 0.14 ± 0.003 | | 0.20 |
| Physical activity (MET) | Normal weight | 23 | 0.22 ± 0.19 | 0.07 | 121,666 | 0.12 ± 0.003 | $<2\times10^{-16}$ | 0.60 |
| | Obesity | 61 | −0.21 ± 0.14 | | 85,012 | −0.20 ± 0.004 | | 0.91 |
| $PRS_{BMI}$ | Normal weight | 28 | −0.64 ± 0.18 | $1.7\times10^{-6}$ | 147,554 | −0.29 ± 0.003 | $<2\times10^{-16}$ | 0.05 |
| | Obesity | 76 | 0.40 ± 0.11 | | 109,216 | 0.37 ± 0.003 | | 0.76 |

| Variable | BMI category | Carriers $N_{total}$ | Carriers N | Carriers % | Carriers P value (normal weight versus obesity) | Noncarriers $N_{total}$ | Noncarriers N | Noncarriers % | Noncarriers P value (normal weight versus obesity) | P value (carrier versus noncarrier) |
|---|---|---|---|---|---|---|---|---|---|---|
| Sex (men) | Normal weight | 28 | 9 | 32% | 0.50 | 147,545 | 51,181 | 35% | $<2\times10^{-16}$ | 0.80 |
| | Obesity | 76 | 33 | 43% | | 109,200 | 52,769 | 48% | | 0.40 |
| Current smokers | Normal weight | 27 | 6 | 22% | 0.55 | 147,082 | 16,477 | 11% | $<2\times10^{-16}$ | 0.11 |
| | Obesity | 76 | 11 | 14% | | 108,621 | 10,491 | 10% | | 0.20 |

*(Continued)*

**Table 1.** (Continued)

| | | | | | | | | | | |
|---|---|---|---|---|---|---|---|---|---|---|
| Physical activity (IPAQ) | Normal weight | 23 | | | 0.09 | 121,666 | | | <2×10⁻¹⁶ | 0.2 |
| | | | Low | 1 | 4% | | | 17,410 | 14% | |
| | | | Moderate | 10 | 43% | | | 49,777 | 41% | |
| | | | High | 12 | 52% | | | 54,479 | 45% | |
| | Obesity | 61 | | | | 85,012 | | | | 0.45 |
| | | | Low | 19 | 31% | | | 22,287 | 26% | |
| | | | Moderate | 14 | 23% | | | 34,184 | 40% | |
| | | | High | 28 | 46% | | | 28,541 | 34% | |
| Comparative body size at age 10 years | Normal weight | 28 | | | 0.01 | 145,241 | | | <2×10⁻¹⁶ | 0.05 |
| | | | Below average | 9 | 32% | | | 57,971 | 40% | |
| | | | Average | 12 | 43% | | | 73,899 | 51% | |
| | | | Above average | 7 | 25% | | | 13,371 | 9% | |
| | Obesity | 75 | | | | 107,073 | | | | 1.0×10⁻⁴ |
| | | | Below average | 13 | 17% | | | 26,759 | 25% | |
| | | | Average | 22 | 29% | | | 51,128 | 48% | |
| | | | Above average | 40 | 53% | | | 29,186 | 27% | |
| Comparative height at age 10 years | Normal weight | 28 | | | 0.30 | 145,288 | | | 1.6×10⁻⁸ | 0.20 |
| | | | Below average | 2 | 7% | | | 30,541 | 21% | |
| | | | Average | 18 | 64% | | | 77,041 | 53% | |
| | | | Above average | 8 | 29% | | | 37,706 | 26% | |
| | Obesity | 76 | | | | 107,060 | | | | 0.98 |
| | | | Below average | 15 | 20% | | | 20,737 | 19% | |
| | | | Average | 41 | 54% | | | 58,656 | 55% | |
| | | | Above average | 20 | 26% | | | 27,667 | 26% | |

Data for continuous traits are expressed in SD scores (i.e., we calculated residuals after adjusting for age and the first 10 principal components in men and women, followed by inverse normal transformation to a distribution with mean of 0 and SD of 1).

BMI, body mass index; FFMI, fat-free mass index; IPAQ, International Physical Activity Questionnaire; MET, metabolic equivalent minutes; $PRS_{BMI}$, polygenic risk score for BMI; SE, standard error; WHR, waist-to-hip ratio.

carrier had undergone weight loss surgery and was removed from further analyses, leaving 182 carriers. Because we restricted our analyses to high-impact mutations, the average age- and sex-adjusted BMI of carriers (mean ± SD = 29.9 ± 5.1 kg/m$^2$; or 0.51 ± 0.08 SD scores [SDSs]) was substantially higher than that of noncarriers (27.4 ± 4.7 kg/m$^2$; −0.0002 ± 0.001 SDS). Furthermore, although most carriers suffered from obesity ($N_{OB}$ = 76, 42%) or overweight ($N_{OW}$ = 78; 43%), 28 (15%) carriers were of normal weight, defying their genetic risk. We next investigated what sets these 28 carriers of normal weight apart from carriers with obesity and from noncarriers of normal weight.

## Body composition in *MC4R* mutations carriers compared to noncarriers

Body composition of individuals of normal weight did not differ between carriers and noncarriers (Table 1; S6 Table). However, among individuals with obesity, carriers had a significantly higher BMI (0.7 kg/m$^2$ equivalent to 2 kg for a 1.7-m-tall person, $P$ = 0.03) than noncarriers, which may be driven by a higher FFMI (0.7 kg/m$^2$, $P$ = 0.001). Carriers with obesity tended to be somewhat taller than noncarriers, but this difference did not reach significance (1.8 cm, $P$ = 0.09) (Table 1; S6 Table).

## Normal-weight *MC4R* mutation carriers compared to carriers with obesity

Although at birth, carriers of normal weight already tended ($P$ = 0.05) to be lighter than carriers with obesity, the difference in body size became more apparent by age 10 ($P$ = 0.01). Carriers of normal weight reported more often (75%) to be below average or of average body size at age 10 (compared with peers) than carriers with obesity (46%) (Table 1), suggesting that carriers of normal weight may have been able to resist weight gain already at a young age.

We next examined the role of innate (genetic) and environmental/lifestyle (nongenetic) factors to determine which compensatory mechanisms contribute to the ability of some carriers to remain of normal weight.

## Low polygenic susceptibility protects MC4R mutation carriers from obesity

We assessed people's overall genetic susceptibility to obesity using a polygenic risk score ($PRS_{BMI}$) based on the BMI association of common variants across the genome. We found that the $PRS_{BMI}$ of normal-weight carriers was more than 1 SDS (corresponding to 1.34 kg/m$^2$ or approximately 4 kg in body weight for a 1.7-m-tall person) lower than that of carriers with obesity ($P < 1.7{\times}10^{-6}$) (Table 1, Fig 3). This difference was only slightly attenuated after additionally adjusting for physical activity (metabolic equivalent minutes [MET]), smoking behavior, and TDI (S7 Table). Even among individuals of normal weight, carriers had a 0.35 SDS (0.45 kg/m$^2$ or approximately 1.3 kg in body weight) lower $PRS_{BMI}$ than noncarriers ($P$ = 0.05) (Table 1, Fig 3, S3 Fig). These observations suggest that carriers of normal weight offset their increased obesity risk caused by *MC4R* mutations, at least in part, because of a low polygenic risk.

Carriers of normal weight tended to be exposed to a healthier environment: less deprivation ($P$ = 0.05) and more physical activity ($P$ = 0.07) compared to carriers with obesity (Table 1). However, because these data were collected cross-sectionally, we cannot determine whether the healthier environment is the cause or consequence of being normal weight.

## Polygenic risk impacts the obesity-increasing effect of *MC4R* mutations

We next assessed the extent to which people's polygenic risk ($PRS_{BMI}$) affects BMI and obesity risk among carriers and noncarriers (Table 2, Fig 4). Compared with noncarriers in the

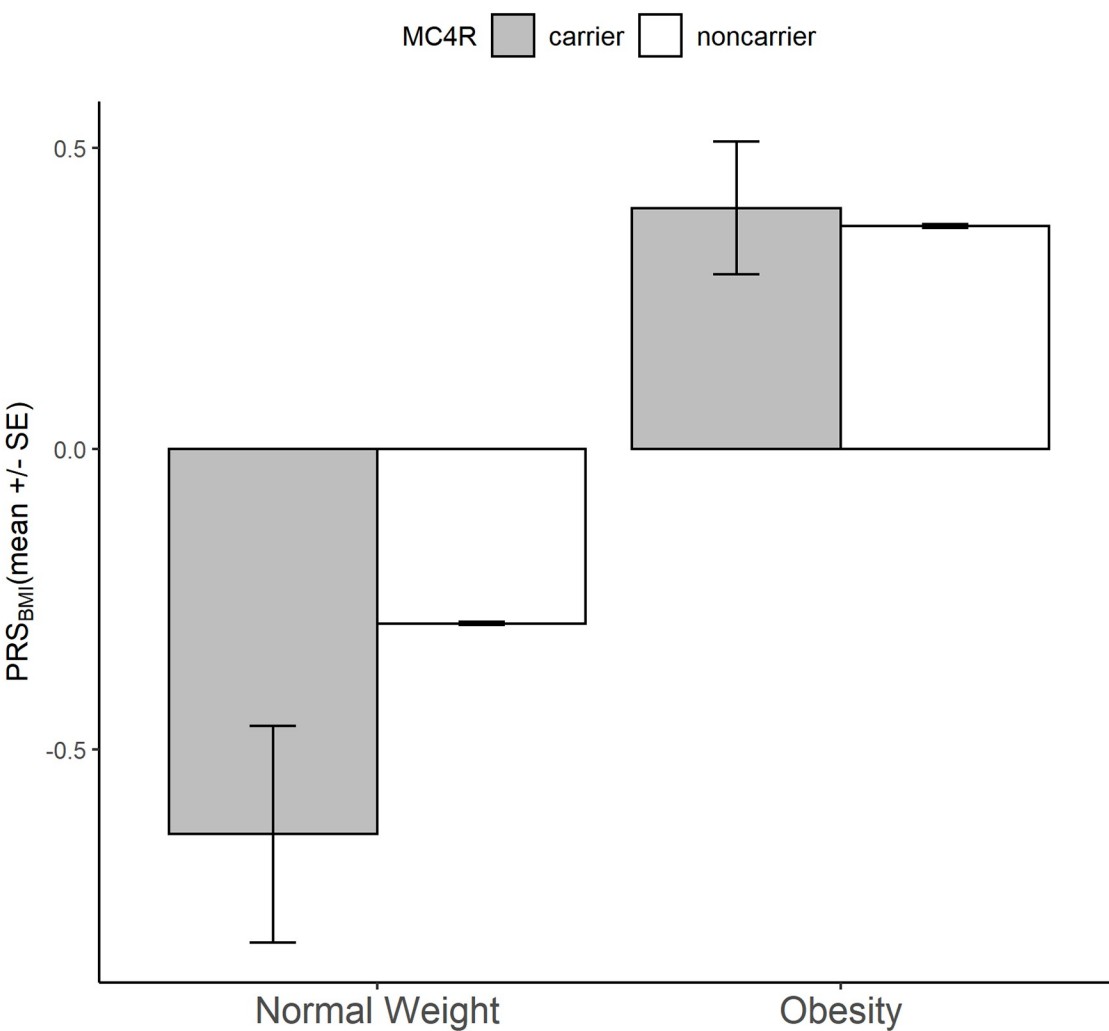

**Fig 3. Polygenic risk (PRS$_{BMI}$) in *MC4R* carriers and noncarriers.** Mean standardized PRS$_{BMI}$ (*y*-axis) for carriers and noncarriers of normal weight and with obesity, respectively. MC4R, melanocortin 4 receptor; PRS$_{BMI}$, polygenic risk score for BMI; SE, standard error.

bottom PRS$_{BMI}$ quartile (lowest risk), carriers in the top PRS$_{BMI}$ quartile (highest risk) have a 9.7-fold increased odds of having obesity (95% CI, 5.6–16.9). The odds of having obesity for noncarriers in the top PRS$_{BMI}$ quartile (medium risk) is 3.7 (95% CI, 3.6–3.7), and for carriers in the bottom PRS$_{BMI}$ quartile (medium risk) is 2.12-fold higher (95% CI, 1.1–4.3), compared with noncarriers in the bottom PRS$_{BM}$ (the lowest risk) (Table 2). Furthermore, the average

**Table 2. Risk of obesity among MC4R carriers and noncarriers with high and low polygenic risk.**

| PRS$_{BMI}$ | *MC4R* carrier status | Risk group | N | Risk of obesity OR (95% CI) | *P* value |
|---|---|---|---|---|---|
| Low (bottom quartile) | noncarrier | Reference | 113,661 | 1 | |
| | carrier | Medium | 44 | 2.2 (1.1–4.3) | 0.028 |
| High (top quartile) | noncarrier | Medium | 113,260 | 3.7 (3.6–3.7) | $2.0\times10^{-16}$ |
| | carrier | High | 52 | 9.7 (5.6–16.9) | $1.2\times10^{-15}$ |

*MC4R*, melanocortin 4 receptor gene; OR, odds ratio; PRS$_{BMI}$, polygenic risk score for body mass index.

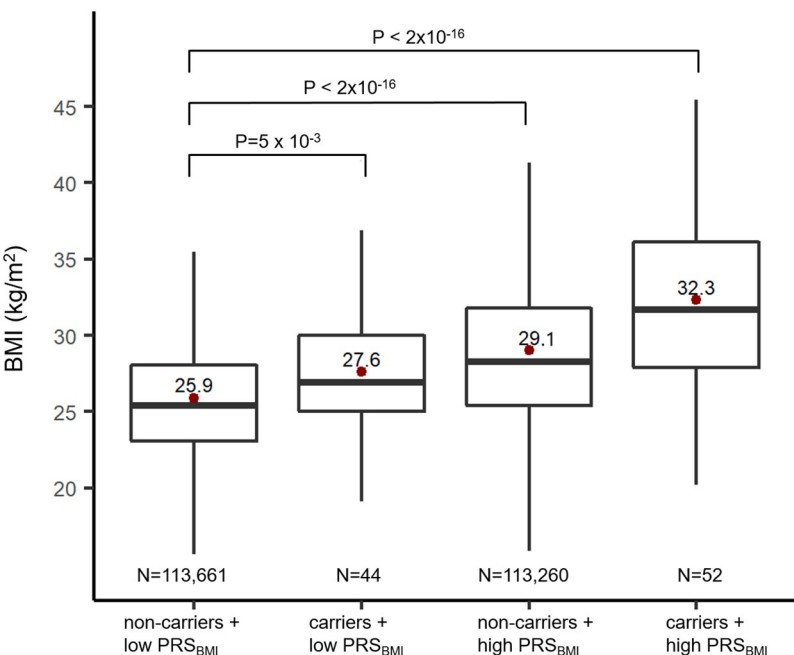

**Fig 4. BMI among *MC4R* carriers and noncarriers in the top and bottom PRS$_{BMI}$ quartiles.** The box represents the median and interquartile range; the red dot is the mean. Low/high PRS$_{BMI}$: bottom/top quartile, high PRS$_{BMI}$. *MC4R*, melanocortin 4 receptor gene; PRS$_{BMI}$, polygenic risk score for body mass index.

BMI difference between the lowest risk (bottom PRS$_{BMI}$, noncarriers) and the highest risk group (top PRS$_{BMI}$, carriers) was 6.4 kg/m$^2$ (or 18.5 kg in body weight for a 1.7-m-tall person, or an approximately 25% difference) (Fig 4). Carrier status was associated with a 1.7 kg/m$^2$ (or 4.9 kg, 6.3%) higher BMI among the low PRS$_{BMI}$ group and with a 3.2 kg/m$^2$ (or 9.2 kg, 11%) higher BMI among the high PRS$_{BMI}$ group.

## Discussion

In contrast to previous, mainly small-scale, often case-focused studies that reported mutations in *MC4R* claimed to cause severe early onset obesity, we leveraged data from over 450,000 individuals and conducted one of the largest studies to validate *MC4R* mutations to date. Of the 59 *MC4R* mutations available in the UK Biobank, only 11 had an impact on obesity risk. Although carriers of these 11 *MC4R* mutations ($N$ = 182) were overall more likely to have obesity, 15% ($N$ = 28) were of normal weight. In these normal-weight carriers, the obesity-increasing effects of the *MC4R* mutations are offset through—at least in part—a significantly lower polygenic risk compared with the carriers with obesity. In addition, compared with noncarriers with a low polygenic risk, mutation carriers with a low polygenic risk have a 2-fold increased risk of obesity and weigh approximately 6% more, whereas carriers with a high polygenic risk have a 9-fold increased risk and weigh approximately 25% more. Taken together, a low polygenic susceptibility to obesity seems to attenuate the impact of pathogenic mutations in *MC4R*.

Studies reporting on *MC4R* mutations that cause severe and early onset obesity have typically been small and case-focused studies and may therefore have overestimated their impact. With the availability of large-scale population studies, such as the UK Biobank, it has become possible to assess these mutations' impact on obesity at a population level. So far, one other study has examined the impact of *MC4R* variants on obesity in the full UK Biobank population

using genotype data [33]. In their study, 61 *MC4R* variants (MAF < 2%) were functionally characterized; 47 variants resulted in LoF, 9 in gain-of-function (GoF), and 5 had no clear functional impact. All of the high-impact mutations identified in the current study that overlapped with the 61 variants were among the ones with the largest impact on MC4R function, based on their low-to-no β-arrestin recruitment [33]. In a second study, *MC4R* was sequenced in a subset of 6,547 UK Biobank participants, identifying 23 protein-altering mutations of which 54 carriers had higher BMI than the noncarriers, a difference that was even more pronounced when only the pathogenic mutations were considered [34], in line with our observations. However, neither study examined the penetrance of the observed *MC4R* mutations, nor did they examine the role of the PRS among carriers.

Consistent with reports on putative causal variants for other diseases [28,35–37], the majority (84%) of mutations previously claimed to cause severe and early onset obesity do not show convincing penetrance or association with obesity in approximately 450,000 individuals of the UK Biobank. For just 11 (19%) of the 59 mutations, the prevalence of obesity among mutation carriers (42%) was substantially higher than among noncarriers (25%) and mutations carriers had a 3.5-fold higher odds of suffering from obesity than noncarriers. Obesity risk for the remaining 48 *MC4R* mutations was not different between carriers and noncarriers, even though for more than a third of these 48 mutations, in vitro analyses have shown evidence of functional implications (S4 and S5 Tables). In fact, for 25 of these 48 mutations, the minor alleles were even more often observed in individuals who did not have obesity and thus associated with lower risk of obesity. Thus, our results suggest that current variant annotations and in vitro functional analyses provide only partial information on the impact of mutations. We show that many of the *MC4R* mutations previously reported to cause severe early onset obesity may not be as pathogenic as previously thought.

We focused our subsequent analyses on 11 *MC4R* mutations that did have an impact on obesity in the approximately 450,000 UK Biobank individuals of European ancestry. Despite the fact that carriers of these 11 mutations were at a substantially higher risk of obesity than noncarriers and that all but one of these mutations had been shown to have functional implications, 28 mutation carriers were of normal weight. Understanding why some mutation carriers remain of normal weight can help elucidate protective mechanisms and provide new targets for treatment and prevention, as has been shown for other disease outcomes [38–41].

*MC4R* mutations affect body weight from an early age onward [6,42,43]. Nevertheless, based on self-reported life-course data, the carriers of normal weight seem to have been of normal weight from a young age, and likewise, the carriers with obesity seem to have been larger than average since childhood. This suggests that the genetic and/or nongenetic mechanisms that counteract the obesity-increasing effects of *MC4R* mutations in carriers of normal weight act throughout the life course. In adulthood, body composition of normal weight carriers was the same as for normal-weight noncarriers. However, carriers with obesity had a higher BMI (likely because of more lean mass) than noncarriers with obesity, consistent with was has been observed for *MC4R* mutations carriers before [6].

Most interestingly, we show that some carriers seem able to counteract the obesity-increasing effects of the *MC4R* mutations they carried and remain of normal weight because—at least in part—of their substantially lower polygenic susceptibility (>1 SD in $PRS_{BMI}$) than carriers with obesity. The normal-weight carriers' polygenic susceptibility was even lower than that of normal-weight noncarriers, suggesting that an "overcompensation" was needed for them to be of normal weight. Furthermore, the impact of the obesity-increasing *MC4R* mutations is substantially attenuated in individuals with a low polygenic susceptibility compared with those with a high polygenic susceptibility. For example, the difference in BMI between carriers and noncarriers is 1.7 kg/m$^2$ (or 4.9 kg, 6%) among individuals with a low polygenic susceptibility

and almost double (3.2 kg/m$^2$ or 9.2 kg, 11%) among those with a high polygenic susceptibility. Carriers with a low polygenic risk were even leaner (1.5 kg/m$^2$ or 4.3 kg, 6%) than noncarriers with a high polygenic risk, whereas the difference was largest (6.4 kg/m$^2$ or 18.5 kg, 25%) between noncarriers with low polygenic risk and carriers with a high polygenic risk. These observations illustrate that both *MC4R* mutations and polygenic susceptibility contribute to people's body weight and that one can attenuate or exacerbate the other.

However, differences in polygenic susceptibility do not fully explain the difference in BMI between carriers of normal weight and carriers with obesity; other genetic and nongenetic mechanisms are likely implicated. Recent studies have speculated that variable penetrance of functional mutations may be due to the fact that in heterozygous carriers, gene function is "rescued" by the "healthy" allele [44,45]. Determining whether this is the case in carriers of normal weight and not in carriers with obesity would require in vitro functional follow up. Besides genetic factors, nongenetic factors may also partially explain why some carriers are able to remain of normal weight, as weight loss surgery [46–48], lifestyle interventions [49,50], and weight loss medication [51] have been shown to affect the weight of *MC4R* mutation carriers to some extent. We found that carriers of normal weight, compared with carriers with obesity, reported less material deprivation, a known contributor to obesity risk [52]. Because the data on deprivation were collected cross-sectionally, we were not able determine whether less deprivation is causally related to lower body weight.

An accurate estimation of penetrance and effect of mutations on disease is important for clinical genetic testing. A correct diagnosis, specifically in children with severe and early onset obesity, allows implementation of tailored treatment and care. We acknowledge that the UK Biobank population is healthier and less deprived than the general UK population [53,54]; i.e., as individuals with obesity may have been less inclined to participate, penetrance and effect may have been somewhat underestimated. Furthermore, only genotype-array data were available for all participants [18], which captured roughly half of all *MC4R* mutations previously reported to cause severe early onset obesity. Although the accuracy of genotype data for very rare mutations has been a concern [28], we implemented stringent quality control measures to remove low-quality mutations. Sensitivity analyses in which we excluded additional mutations with potential quality concerns continue to support our primary findings. The exome sequencing dataset currently constitutes approximately 10% of the total UK Biobank population and is still too small to examine the rare (MAF < 0.1%) mutations of interest [55].

Taken together, our findings suggest that, in addition to existing case-focused studies and in vitro functional analyses, large-scale population data are required to more accurately assess the impact of *MC4R* mutations (and potentially others) on severe early onset obesity. In addition, we show that the impact of even highly penetrant *MC4R* mutations may be at least partially mitigated by a low polygenic susceptibility to obesity and potentially healthier environments and behaviors, blurring the lines between monogenic and polygenic forms of obesity.

## Supporting information

**S1 Text. Supplementary methods.**
(PDF)

**S1 Fig. Sensitivity analysis: Standardized PRS$_{BMI}$ values in carriers and noncarriers of normal weight versus obesity after excluding rs1367004987, Affx-89021050, and rs775382722 from the high-impact variants.** PRS$_{BMI}$, polygenic risk score for BMI.
(TIF)

**S2 Fig. Sensitivity analysis: BMI among carriers and noncarriers in the top and bottom quartiles of PRS$_{BMI}$ after excluding rs1367004987, Affx-89021050, and rs775382722 from the high-impact variants.** BMI, body mass index; PRS$_{BMI}$, polygenic risk score for BMI. (TIF)

**S3 Fig. Density plots of PRS$_{BMI}$ for *MC4R* mutation carriers and noncarriers with obesity and of normal weight, respectively.** *MC4R*, melanocortin 4 receptor gene; PRS$_{BMI}$, polygenic risk score for BMI. (TIF)

**S1 Table. Descriptive characteristics of 451,508 individuals of European Ancestry from the UK Biobank.** (XLSX)

**S2 Table. Quality of 69 genotyped mutations in *MC4R* in UK Biobank participants of European ancestry.** The sequencing dataset is based on the European subset only because of allele frequency differences for some variants between the European and other populations, such as the African American population. It is also based on the same subset that was analyzed in our analyses of the genotyping data. *Refers to the number of individuals who were carriers in the genotyped dataset but noncarriers in the sequencing dataset. **Refers to the number of individuals who were carriers in the sequenced dataset but noncarriers in the genotyping dataset. *** FFP = high (>25%) false positive proportion; NFP = high (>25%) false negative proportion. *MC4R*, melanocortin 4 receptor gene. (XLSX)

**S3 Table. Sensitivity analysis: Comparison between carriers and noncarriers of normal weight versus obesity after removing rs1367004987, rs775382722, and Affx-89021050.** Values are expressed in SD scores (i.e., we calculated residuals after adjusting for age and the first 10 PC in men and women, followed by inverse normal transformation to a distribution with mean of 0 and SD of 1). Cochran–Armitage test for trend was used to compare carriers and noncarriers as well as individuals of normal weight and with obesity for IPAQ, comparative body size at age 10 years and comparative height at age 10 years. *P* values are reported for adjusted means for age, sex, and 10 PCs. IPAQ, International Physical Activity Questionnaire; MET, metabolic equivalent minutes; PC, principle component; PRS$_{BMI}$, standardized scores of the polygenic risk score of BMI with mean 0 and SD of 1; SD, standard deviation. (XLSX)

**S4 Table. The number of carriers for each mutation stratified by BMI category and their impact on obesity.** *Because there are mutations for which there were no normal-weight carriers, we added a constant ("+1") to the number of carriers with obesity (numerator) and to the number of carriers of normal weight (denominator) to allow deriving "proxy" ORs. **Functional evidence refers to existing literature that demonstrated that the mutation leads to lower expression, impaired signaling or loss of function. "Ambiguous" refers to mutations that had conflicting reports of either supporting or refuting downstream effects or for which the loss of function effects of the mutation were not clear. BMI, body mass index; OR, odds ratio. (XLSX)

**S5 Table. Annotation of the 59 *MC4R* mutations.** Functional evidence refers to existing literature that demonstrated that the mutation leads to lower expression, impaired signaling or loss of function. "Ambiguous" refers to mutations that had conflicting reports of either

supporting or refuting downstream effects or for which the loss of function effects of the mutation were not clear. *MC4R*, melanocortin 4 receptor gene.
(XLSX)

**S6 Table. Comparison between carriers and noncarriers of normal weight versus obesity.**
Values for anthropometry and lifestyle factors are inferred from the standardized scores
(Table 1) that were adjusted for age, PCs in men and women separately). PC, principle component.
(XLSX)

**S7 Table. Comparison of PRS$_{BMI}$ between carriers and noncarriers stratified by BMI category (in adjusted standardized scores).** *P* values are reported for adjusted means. Model 1 is the adjusted mean of the PRS after including age, sex, and the first 10 principal components in the model. Model 2 is the adjusted mean of the PRS after adding in addition to the covariates in model 1, MET scores, current smoking, and the Townsend Deprivation Index. BMI, body mass index; PRS$_{BMI}$, polygenic risk score for BMI with mean 0 and SD of 1.
(XLSX)

**S1 STROBE Checklist. STROBE, Strengthening The Reporting of OBservational Studies in Epidemiology.**
(DOCX)

## Author Contributions

**Conceptualization:** Nathalie Chami, Ruth J. F. Loos.

**Data curation:** Nathalie Chami, Michael Preuss, Ryan W. Walker, Arden Moscati.

**Formal analysis:** Nathalie Chami, Michael Preuss, Ryan W. Walker, Arden Moscati.

**Funding acquisition:** Ruth J. F. Loos.

**Investigation:** Nathalie Chami, Ruth J. F. Loos.

**Methodology:** Nathalie Chami, Michael Preuss, Ruth J. F. Loos.

**Project administration:** Ruth J. F. Loos.

**Resources:** Arden Moscati, Ruth J. F. Loos.

**Supervision:** Ruth J. F. Loos.

**Validation:** Nathalie Chami.

**Writing – original draft:** Nathalie Chami, Ruth J. F. Loos.

**Writing – review & editing:** Nathalie Chami, Michael Preuss, Ryan W. Walker, Arden Moscati, Ruth J. F. Loos.

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
