## [Editor Report · Decision Letter 0]

7 Feb 2020

Dear Dr Loos, 

Thank you for submitting your manuscript entitled "Low polygenic risk attenuates the obesity-increasing effects of pathogenic mutations in MC4R" for consideration by PLOS Medicine.

Your manuscript has now been evaluated by the PLOS Medicine editorial staff [as well as by an academic editor with relevant expertise] and I am writing to let you know that we would like to send your submission out for external peer review.

Kind regards,

Adya Misra, PhD,

Senior Editor

PLOS Medicine

---

## [Decision Letter · Decision Letter 1]

9 Apr 2020

Dear Dr. Loos,

Thank you very much for submitting your manuscript "Low polygenic risk attenuates the obesity-increasing effects of pathogenic mutations in MC4R" (PMEDICINE-D-20-00082R1) for consideration at PLOS Medicine. 

[LINK]

In light of these reviews, I am afraid that we will not be able to accept the manuscript for publication in the journal in its current form, but we would like to consider a revised version that addresses the reviewers' and editors' comments. Obviously we cannot make any decision about publication until we have seen the revised manuscript and your response, and we plan to seek re-review by one or more of the reviewers. 

We expect to receive your revised manuscript by Apr 30 2020 11:59PM. Please email us (plosmedicine@plos.org) if you have any questions or concerns.

We look forward to receiving your revised manuscript. 

Sincerely,

Adya Misra, PhD

Senior Editor 

PLOS Medicine

plosmedicine.org

Title-Please revise your title according to PLOS Medicine's style. Your title must be nondeclarative and not a question. It should begin with main concept if possible. "Effect of" should be used only if causality can be inferred, i.e., for an RCT. Please place the study design ("A randomized controlled trial," "A retrospective study," "A modelling study," etc.) in the subtitle (ie, after a colon).

Abstract

Background- is it monogenic forms of obesity? Please specify this point if so

Background-please clearly highlight the aim of your study

Format-Please structure your abstract using the PLOS Medicine headings (Background, Methods and Findings, Conclusions).

Please combine the Methods and Findings sections into one section, “Methods and findings”.

Please include the study design, population and setting, number of participants, years during which the study took place, length of follow up, and main outcome measures

Please provide 95% confidence intervals along with p values

Please clarify this sentence “Normal weight carriers more often reported that, already at age 10y, they were thinner/average (72%) compared to obese carriers (48%) (P=0.02)”. From the language it appears as though the weight was self reported which should be clarified in the methods section of the abstract. Also it is not clear if the measure is BMI or body composition, Please revise as needed.

Abstract Conclusions:

* Please address the study implications without overreaching what can be concluded from the data; the phrase "In this study, we observed ..." may be useful.

* Please interpret the study based on the results presented in the abstract, emphasizing what is new without overstating your conclusions.

* Please avoid vague statements such as "these results have major implications for policy/clinical care". Mention only specific implications substantiated by the results.

* Please avoid assertions of primacy ("We report for the first time....")

Author summary

Introduction

References must be in Vancouver style and provided within square brackets please. 

Paragraph 3 on page 4- I assume this is still MC4R mutations so it might be better to mention this in the first two sentences

Please move methods to just after Introduction on page 5

Results

Please simplify this sentence: “For 10 (91%) of the 11 mutations, there was evidence that the mutation impaired MC4R function and/or led to reduced activity, based on functional characterization (Tables S3 and S4), which was significantly more often than for the remaining 48 mutations (P = 0.0006) for which we found evidence for only 17 (33%)”. You may consider splitting up the sentence as it is currently hard to follow.

Please report p values of up to two decimal places

Please provide “(no individuals carried more than one mutation)” as a separate sentence. 

Page 7 please introduce FFMI on first view

Discussion

Please rephrase “beating their genetic odds” on page 8. The same goes for “counteract the obesity-increasing effects of MC4R mutations”. 

Recommend revising instances “normal phenotype” to non-obese or similar, to avoid any stigmatising labels. 

Please avoid assertions of primacy such as “we show for the first time” by adding “to our knowledge”

You mention sensitivity analyses here but not in the results? Please provide these as SI files as needed to support your findings

Limitations of the UK biobank cohort and your methodology (specifically self reported weight) more generally must be outlined in further detail.

Page 22- please revise the last sentence containing multiple instances of (R,2013)

Please ensure that the study is reported according to a appropriate reporting guideline (GRIPS? Or STROBE), and include the completed checklist as Supporting Information. When completing the checklist, please use section and paragraph numbers, rather than page numbers. Please add the following statement, or similar, to the Methods: "This study is reported as per the xxxx guideline (S1 Checklist)."

Please report your study according to the relevant guideline, which can be found here: http://www.equator-network.org/

Comments from the reviewers:

Reviewer #1: The authors report the findings investigating the effect of having MC4R mutations on differences in BMI (based on BMI) on between carriers and non-carriers in the UK Bio Bank. They found that 11 MC4R mutations have a high penetrance and that low polygenic risk score is protective of having increased risk of obesity in carriers of MC4R mutations. This study was similar to a recent study (the authors have cited as well) in Cell (https://www.sciencedirect.com/science/article/pii/S0092867419302909?via%3Dihub) using UK Biobank (though that previous study did extend their study to validate in four other datasets as well as explore long-term effects on cardio-metabolic conditions). The genetic risk score from this study mirrored similar size and composition of the previous (> 300,000 SNPs), but with similar findings. This current study does explore BMI between carrier and non-carriers specifically and the results offer some insight in the implications of the protective effects of having low polygenic risk score. 

The statistical methods are sound, constructions of the polygenic risk score followed conventional design using linear additive models, with appropriate quality control and principal components to account for European ancestry. The the environmental variables also seem to provide some insight into the potential effect modification. 

One aspect is that I am not sure the second aim of the study was fully answered on why some individuals who carry these mutations are able to remain of normal weight - largely the results simply report the observed differences between groups and does not investigate the underlying effect modification between environmental factors in each group. For instance, on page 7, the authors state that environmental/lifestyle (non-genetic factors were examined). They have provided some interesting results in on Table 1 comparing lifestyle factors stratifying by carriers and non-carriers. What I find interesting is that there is some definite effect modification by lifestyle factors here. For instance, physical activity (MET and IPAQ) are not different between obese and normal weight carriers but is significantly different in normal and obese non-carriers. Also other demographic attributes like sex and height look to have important differences in association between normal and obese in carriers vs non-carriers. Could the authors elaborate further on whether environmental factors looks like they might have a pretty strong modifying effect - potentially even more so than genetic susceptibility and whether there were any interactions that could be explore in the analysis between gene and environment. 

Some other minor comments: 

Selection of 11 mutations: Authors should rationalise why >= 30% penetrance and >= 2 OR was used to define high impact on obesity

Top of Page 8: Physical activity according was not significant here (P = 0.08) in carriers. I think one aspect that has not been mentioned in the text is the impact of the other environmental risk factors which did seem to have different associations between carriers and non-carriers (smoking, physical activity).

Reviewer #2: Chami et al manuscript Low polygenic risk attenuates the obesity-increasing effects of pathogenic mutations in MC4R

The authors have leveraged the large-scale dataset from UKB to understand the relevance of heterozygous mutations in MC4R to the risk of obesity. This is an important area of interest and large population studies such as UKB may provide information on the penetrance and pathogenicity in the relatively healthy populations while compared to the possible ascertainment bias in the small-scale disease focused population. 

Reviewer comments: 

1) The most important limitation of this study is the use of genotype data to assess rare variants without confirmation of the variants. The authors report the use of the tool Evoker (Morris et al 2010) to ascertain the quality of the variants resulting in 59 variants of "good" quality. Evoker is being extended here from its original use for common variants. Based on the use of Evoker lite, Wright et al 2019 have noted that variants below MAF 0.001% are not reliably genotyped with the false positive rates ~60% in data from UKB, while those with MAF > 0.005% was ~20%. It is to be noted that 29 (out of 69) variants have MAF <= 0.001% including some from the 11 considered high penetrance and 59 variants have MAF <= 0.005%. Lotta et al 2019 used the same method for validation of the variants published in their paper, and the authors need to provide further evidence that these variants are true positives before perpetuating this method for ascertaining rare variants further. There is some evidence in the literature of the false-positive results of rare variants in Mendelian genes ascertained from genotyping data. 

In their own comparison of the subset of the sequencing data with the corresponding genotype data, the authors postulate that 28 of the variants were deemed to be of low quality. While this reviewer acknowledges the limitation of scaling the results from analyses of ~10% of the data to the entire cohort. Most studies using exome data also validate the identified variants by one or other method prior to publication. Given the substantial implications of the conclusions of this manuscript, is it prudent to wait till the sequencing data for the entire UKB becomes available prior to making the assertion? If indeed a large number of variants are found to be false positive, would the conclusions still hold? The stochastic nature of such false positive findings makes it difficult to identify true positive amongst the variants noted in this study. It is possible that the conclusions of the study will hold after the due diligence of validation of the variants in which case, this publication will be applauded for accurate paradigm shifting conclusions. The influence of polygenic risk on BMI is not to be underestimated, especially when considering a similar effect in individuals who are not carriers of variants in MC4R. It is just not clear if the low polygenic risk is "protective" if the validity of the variants in MC4R is not established unequivocally. 

2) This manuscript focuses on individuals with overweight/obesity, while completely ignoring the principal of "people first language". People first language has been widely recognized as important for use in academic publications relevant to individuals with obesity (Kyle et al 2014, PMID 24616446, Wittert et al, PMID 26373880 and several others). The authors need to review their language throughout the manuscript, e.g. "normal weight carriers" will be carriers with normal weight, "obese carriers" will be carriers with obesity. "mutation carriers" should be carriers with mutations, "non-MC4R deficient individuals" should be individuals without mutations in MC4R etc. It is critical for scientists to remember that people are more important than pathology, always.

3) In the phenotype review, the authors have included the assessment of anthropometric parameters, but not the diagnoses codes or health status of the individuals. While the population enrolled in UK Biobank is expected to be healthier, for this manuscript, it will be important for the authors to review the ICD codes for the individuals under study to ensure that the "normal" weight was not due to an underlying illness? Further, as the authors are well aware that environmental influences far exceed indices of socioeconomic status and physical activity. How would the authors account for educational status as a measure of the socioeconomic awareness, smoking status and dietary habits in their modeling? Given the extensive phenotype data ascertained by UKB, it is naïve to include only age, sex and PCs derived from genetic data in the models. Additional phenotype data ought to be considered, at minimum for exploration or for sensitivity analysis. 

4) Page 6: "For 10 (91%) of the 11 mutations, there was evidence that the mutation impaired MC4R function and/or led to reduced activity, based on functional characterization (Tables S3

and S4), which was significantly more often than for the remaining 48 mutations (P = 0.0006) for which we found evidence for only 17 (33%)."

in table S3, there are 20 variants that have been reported to have functional effect while the text mentions 17. Which one is correct?

5) Authors need to discuss the limitations of using recall data from 10 years of age for adults recruited at 40-60 years of age, especially with reference to a phenotype such as height and weight and be conservative in their conclusions from such data in the abstract.

6) Page 7: "These observations suggest that normal weight carriers are able to overcome their increased obesity risk due to MC4R mutations, at least in part, thanks to a low polygenic risk." 

Scientific articles should not be using language that glorifies one weight category over the other. 

There are a few other instances where colloquial language is used: "the extent to which people's polygenic risk (PRSBMI) affects BMI and obesity risk among carriers and non-carriers"

Reviewer #3: SUMMARY 

Alterations in the DNA sequence coding for MC4R is considered the most common form of monogenic obesity, but evidences have been accumulated that the penetrance of the different MC4R isoforms is variable. Aim of the present study is to examine in a large population based study (>450000 subjects of European Ancestry from UK biobank) why some carriers of pathogenic mutations remain of normal weight, with the ultimate goal of acquiring novel knowledge on mechanisms underlying body weight control. 

To this end they put in place the following experimental design. 

1. Selection of MC4R high impact variants based on 2 stringent criteria: 1. Penetrance ≥30% of carriers are obese. 2. Obesity risk (OR) defined as follows : provided that in non carriers the ratio obese/normal weight is 0.74, mutations for which this ratio is ≥2. This approach led to the identification of 11 mutations which met both criteria. For 91% of them functional studies indicating an impaired MC4R function were available. The other 48 mutations present in the literature because associated to obesity or known to alter MC4R functions were discarded from further analysis. 

2. Individuals (182) carrying one of the high impact MC4R variants are then stratified according to BMI. 29 of them show normal body weight (NW), while 75 are obese. Non-carriers of high impact mutations are similarly stratified. Pair comparisons were then performed at different levels: normal weight vs obese or carrier versus non carriers. Parameters taken into account are physical parameters, history of body weight, life style, socio-economic conditions. Results indicate that NW carriers were protected against obesity already at age 10, and show better socioeconomic conditions compared to obese carriers. 

3. When polygenic risk susceptibility score (PRS) for obesity is taken into account the carriers with normal weight show a much lower PRS, compared with obese carriers and also a lower PRS when compared to normal weight non carriers, indicating that normal weight carriers defy their obesity risk with a very low PRS.

Their conclusion is that large scale population studies are very important to accurately assess the impact of MC4R mutations on severe obesity and that the line between polygenic and monogenic forms of obesity is not that sharp. 

GENERAL COMMENT 

The genetics of obesity field has been crowded for the past 25 years with case control studies based on small cohorts, which in most cases reported a modest difference between normal weight and obese subjects for a given mutation. This is particularly true in the case of MC4R. Furthermore, criteria on subjects stratification has varied. Overall this has produced a huge set of not organized, not reliable data, and confusing information, which led many scholars in the field to open their papers with a generic sentence stating that monogenic forms of obesity represent 5% of the total obese population and that MC4R variants account for most of these forms. 

Chami and colleagues provide a very timely and important piece of research, which I welcome as an unmet need and a new starting point. Rigorous methods, solid evidences and brilliant discussion characterize this paper. 

Below a few suggestions to make it more clear and usable from a large audience 

DETAILED COMMENTS 

- This study is not easy to read for clinicians and biologist unless they have a strong background in modern genetics and statistics. My recommendation is to provide both graphics and explanations to better guide the reader throughout the experimental design. A scheme putting into evidence the main questions addressed as well as the relative answers should be included. 

- The 11 mutations with high impact are selected based on penetrance (>30%) and obesity risk (increase>2 fold). More in deep and detailed explanations should be provided to explain these criteria which may sound arbitrary otherwise. 

- The last column of table 1 as well as of the supplementary tables indicate the P value when carriers and non carriers are compared. The sample size of these groups differ by various order of magnitude. Is it right to compare groups with so different sample size , i.e 29 versus > 100000? Provide explanations

Reviewer #4: The MC4R is a central player in the leptin-melanocortin pathway, which plays a critical role in the brain control of food intake and body-weight. Genetic disruption of the pathway is known to result in severe obesity. Mutations in the MC4R are no exception, and have been known, since 1998, to be strongly associated with increased body-weight. What is interesting is that while non-synonymous mutations in this receptor are relatively common (compared to super rare conditions), questions remain about its penetrance woth regards to increased body-weight. In this manuscript, Chami and colleagues study the impact of non-synonymous variations in the MC4R gene on body-weight within UK Biobank. They note that about 25% of people carrying loss of function MC4R variants are not obese, and report that their polygenic risk score with regards to BMI can mitigate against carrying an MC4R mutation. This is an important study which tackles the nuance of what was previously thought to be a 'monogenic' cause of obesity. I've got a few issues to raise:

1. The authors do acknowledge that UK Biobank represents a healthier slice of society. Given that deprivation is inversely correlated with health, how does the deprivation index of UK biobank compare to other large population cohort studies?

2. While the carriers with a lower polygenic risk score have their MC4R genetic risk mitigated, does the specific SNP (rs571312) near MC4R play any increase or decreased role in this mitigation?

3. I apologise if this is an ignorant question, but in table 1, what exactly are adjusted standardized scores?

[LINK]

---

## [Decision Letter · Decision Letter 2]

28 May 2020

Dear Dr. Loos,

Thank you very much for re-submitting your manuscript "The role of polygenic susceptibility to obesity among carriers of pathogenic mutations in MC4R" (PMEDICINE-D-20-00082R2) for review by PLOS Medicine.

I have discussed the paper with my colleagues and the academic editor and it was also seen again by xxx reviewers. I am pleased to say that provided the remaining editorial and production issues are dealt with we are planning to accept the paper for publication in the journal.

[LINK]

We expect to receive your revised manuscript by 2nd June 2020. Please email us (plosmedicine@plos.org) if you have any questions or concerns.

We look forward to receiving the revised manuscript by Jun 02 2020 11:59PM. 

Sincerely,

Adya Misra, PhD

Senior Editor 

PLOS Medicine

plosmedicine.org

Requests from Editors:

Title- please include a study descriptor or perhaps add “a UK Biobank cohort study” 

Abstract- please include a limitation of your study design/methodology in the last sentence of the 2methods and findings” section

Some of the author summary requires revision for clarity, for example this sentence is a bit unclear “We observed that, compared to carriers with obesity, carriers of normal weight have a lower

overall genetic susceptibility..”. Please revise this as needed

“To date, more than 650 million adults worldwide are obese” should be revised to “suffer from obesity” or similar. Usage of “obese” throughout should be revised in this way throughout the submission please. There are several instances in the methods/results and I would appreciate these are revised- for instance on page 10, 12, 14. Please check Fig 1 for similar language which must be revised. 

Is there anyway we can substitute healthy weight for normal weight? I am concerned that normal weight adds an element of stigma that can be avoided. I think it is fine to use normal weight when describing various weights in relation to BMI in the methods but not to describe individuals. Please revise this throughout the manuscript. There is also the phrase “average weight”, which should probably be revised? (Page 14)

Tables need revision to remove stigmatising language as well. It is very unusual to see words like “Normal versus obese” in scientific articles. In addition, please revise the use of “plumper” when looking at comparative body size. Can you possibly rephrase to “lower than average”, “average” and “greater than average” in this table? I would suggest the same for heights.

Please revise “The odds of having obesity for non-carriers” for grammar as well as using non stigmatising language

Please ensure bibliography is in Vancouver style

Comments from Reviewers:

Reviewer #1: The authors have done a thorough job responding to comments and revising their manuscript. 

I felt that Figure 1 was particularly helpful in understanding their study design and analysis. Pleased to recommend this for acceptance now. 

Reviewer #2: The authors have addressed many of the concerns raised by the editors and reviewers. 

The reviewer appreciates the repetition of the explanation from text and acknowledges that this study is important to be entered into the literature to allow future studies on the WES data to look at this question again. 

There are some minor concerns that can be addressed easily by the authors:

The "people first language" is significantly improved and now it is up to the editors to take care of the rest. 

Page 8: "BMI, calculated as weight (kg) divided by height (m2) squared,.." This comes across a bit odd: (m2) squared will be m4?. It seems like the authors are trying to account for the units, but the calculation really is height in meters squared? Similar issue in the definition of FFMI. 

Page 17 (Discussion):

"Our data suggests….." The data belongs to the UKB. The authors should note this as "Our analysis suggests….

Next, the statement that a "large number of MC4R mutations……." can be more appropriately restated as "many of the mutations….." It will be prudent to avoid hyperbolic claims, as has already been pointed out by the editor. 

Page 19 (Discussion): 

"In addition,we show that the impact of even highly penetrant MC4R mutations can be mitigated by a low". Please rephrase it to account for the limitations of the study, similar to what has already been done in the previous sections: 

" …….may be at least partially mitigated……"

Reviewer #3: The authors satisfactorily addressed all issues raised during the revision process and greatly improved the quality of the manuscript

Reviewer #4: The authors have responded to all of my concerns.

[LINK]

---

## [Editor Report · Decision Letter 3]

16 Jun 2020

Dear Dr. Loos, 

On behalf of my colleagues and the academic editor, Dr. Karine Clément, I am delighted to inform you that your manuscript entitled "The role of polygenic susceptibility to obesity among carriers of pathogenic mutations in MC4R" (PMEDICINE-D-20-00082R3) has been accepted for publication in PLOS Medicine. 

PRODUCTION PROCESS

PRESS

PROFILE INFORMATION

Thank you again for submitting the manuscript to PLOS Medicine. We look forward to publishing it. 

Best wishes, 

Adya Misra, PhD

Senior Editor 

PLOS Medicine

plosmedicine.org